# Changes in Spirulina’s Physical and Chemical Properties during Submerged and Solid-State Lacto-Fermentation

**DOI:** 10.3390/toxins15010075

**Published:** 2023-01-13

**Authors:** Ernesta Tolpeznikaite, Vadims Bartkevics, Anna Skrastina, Romans Pavlenko, Ernestas Mockus, Egle Zokaityte, Vytaute Starkute, Dovile Klupsaite, Romas Ruibys, João Miguel Rocha, Antonello Santini, Elena Bartkiene

**Affiliations:** 1Institute of Animal Rearing Technologies, Faculty of Animal Sciences, Lithuanian University of Health Sciences, LT-44307 Kaunas, Lithuania; 2Institute of Food Safety, Animal Health and Environment “BIOR”, Zemgales Priekšpilsēta, LV-1076 Riga, Latvia; 3Department of Food Safety and Quality, Faculty of Veterinary, Lithuanian University of Health Sciences, LT-44307 Kaunas, Lithuania; 4Institute of Agricultural and Food Sciences, Agriculture Academy, Vytautas Magnus University, LT-44244 Kaunas, Lithuania; 5LEPABE—Laboratory for Process Engineering, Environment, Biotechnology and Energy, Faculty of Engineering, University of Porto, s/n, 4200-465 Porto, Portugal; 6ALiCE—Associate Laboratory in Chemical Engineering, Faculty of Engineering, University of Porto, s/n, 4200-465 Porto, Portugal; 7Department of Pharmacy, University of Napoli Federico II, Via D. Montesano 49, 80131 Napoli, Italy

**Keywords:** Spirulina, fermentation, lactic acid bacteria, gamma-aminobutyric acid, biogenic amines

## Abstract

The aim of this study was to select a lactic acid bacteria (LAB) strain for bio-conversion of Spirulina, a cyanobacteria (“blue-green algae”), into an ingredient with a high concentration of gamma-aminobutyric acid (GABA) for human and animal nutrition. For this purpose, ten different LAB strains and two different fermentation conditions (SMF (submerged) and SSF (solid state fermentation)) were tested. In addition, the concentrations of fatty acids (FA) and biogenic amines (BA) in Spirulina samples were evaluated. It was established that Spirulina is a suitable substrate for fermentation, and the lowest pH value (4.10) was obtained in the 48 h SSF with *Levilactobacillus brevis*. The main FA in Spirulina were methyl palmitate, methyl linoleate and gamma-linolenic acid methyl ester. Fermentation conditions were a key factor toward glutamic acid concentration in Spirulina, and the highest concentration of GABA (2395.9 mg/kg) was found in 48 h SSF with *Lacticaseibacillus paracasei* samples. However, a significant correlation was found between BA and GABA concentrations, and the main BA in fermented Spirulina samples were putrescine and spermidine. Finally, the samples in which the highest GABA concentrations were found also displayed the highest content of BA. For this reason, not only the concentration of functional compounds in the end-product must be controlled, but also non-desirable substances, because both of these compounds are produced through similar metabolic pathways of the decarboxylation of amino acids.

## 1. Introduction

Currently, a healthy lifestyle is very popular, and the practice of balanced diets—including the consumption of functional compounds—is of great interest to both humans and animals. Gamma-aminobutyric acid (GABA) is a functional compound that can be obtained through the decarboxylation of L-glutamate by the glutamate decarboxylase intracellular enzyme [1]. It has been confirmed that GABA can be synthetized by some microorganisms, including yeasts, fungi and bacteria [2,3,4]. However, to ensure efficient synthesis of GABA, a selection of the substrate is required, which should contain its precursors, as well as the appropriate microorganisms for the decarboxylation enzymatic process. Spirulina, which belong to the *Cyanobacteria* class (cyanobacteria) [5,6,7] have a significant content of GABA precursors [8]. In addition, these prokaryotic microalgae are commonly used as a functional food and feed material [5,9], because of their wide range of pharmacological activities [10], including amelioration of heavy metals and pesticide toxicity [11].

In this study, we hypothesized that the functional value of Spirulina (*Arthrospira platensis*) can be increased, by using biomass fermentation with selected lactic acid bacteria (LAB) strains. It was reported that the genus *Lactobacillus* and other cocci LAB have abundant GABA-producing species, including *Levilactobacillus brevis* [4,12,13,14,15,16,17,18], *Lactobacillus buchneri* [19,20], *Lactobacillus delbrueckii* subsp. *bulgaricus* [16,21], *Lactobacillus fermentum* [22,23], *Lactobacillus helveticus* [4,24], *Lacticaseibacillus paracasei* [16,25] and *Lactiplantibacillus plantarum* [4,16,25,26], among others. In this study, *Lactiplantibacillus plantarum* No. 122, *Lacticaseibacillus casei* No. 210, *Lactobacillus curvatus* No. 51, *Lacticaseibacillus paracasei* No. 244, *Lactobacillus coryniformis* No. 71, *Pediococcus pentosaceus* No. 183, *Levilactobacillus brevis* No. 173, *Pediococcus acidilactici* No. 29, *Leuconostoc mesenteroides* No. 225 and *Liquorilactobacillus uvarum* No. 245 strains were tested for Spirulina bioconversion. These LAB strains showed previously desirable antimicrobial and antifungal properties [27]. In addition to antimicrobial properties, some of the LAB can degrade mycotoxins [28], as well as display probiotic traits [29,30]. Nevertheless, and as expected, the metabolic activities of LAB are strongly correlated with environmental and processing factors, including the chemical composition of the substrates, moisture content, fermentation temperature and time, pH, buffer capacity, etc.

Currently, the industry is turning to more sustainable technologies. For this reason, biotechnological processes are also changing to meet sustainability requirements. The fermentation process can be performed in liquid as well as in solid state conditions. Solid-state fermentation (SSF) is a microbial process occurring mostly on the surface of solid materials that have the property to absorb or contain water, with or without soluble nutrients [31,32]. Moreover, SSF is often known to reduce global costs in comparison to liquid fermentation [33]. The low water volume in SSF has a large impact on the economy of the process mainly due to smaller bioreactor size, reduced downstream processing, lower sterilization costs, etc. Furthermore, many SSF processes focus on the utilization of cheap agri-industrial byproducts as culture media [32,34]. Submerged fermentation (SMF) is a very well-known methodology in the scientific literature, while SSF occupies a very small but emerging space in biotechnology [35]. In this study, we also hypothesized that the same microorganisms can be used in both fermentation techniques, but the results will differ due to the enormous differences in processing conditions.

Taking into consideration that LAB can excrete decarboxylases, the decarboxylation process can lead to the formation of desirable (e.g., GABA formation) and/or undesirable (e.g., biogenic amines (BA)) metabolites. Most of the BA are classified as non-desirable compounds, except for beta-phenylethylamine (*β*-PEA), which is attributed to neurotransmitters [36,37]. Beta-phenylethylamine is a well-known and widespread endogenous neuroactive trace amine found throughout the central nervous system in humans [37]. This neurotransmitter modifies the release and the response to dopamine, norepinephrine, acetylcholine and GABA [38].

Moreover, during fermentation, various changes can be obtained, including that which concerns the bioconversion of lipids as well as once LAB can perform FA isomerization, hydration, dehydration and saturation in fermentable substrates [39].

The aim of this study was to select the most appropriate LAB strains for the bio-conversion of Spirulina into an ingredient with a high concentration of GABA to be potentially used in human and animal nutrition. For this purpose, various LAB strains (*Lactiplantibacillus plantarum* No. 122; *Lacticaseibacillus casei* No. 210; *Lactobacillus curvatus* No. 51; *Lacticaseibacillus paracasei* No. 244; *Lactobacillus coryniformis* No. 71; *Pediococcus pentosaceus* No. 183; *Levilactobacillus brevis* No. 173; *Pediococcus acidilactici* No. 29; *Leuconostoc mesenteroides* No. 225; *Liquorilactobacillus uvarum* No. 245) and fermentation conditions (SMF and SSF) were investigated. Taking into consideration the complexity of the changes occurring during the fermentation processes, FA and BA profiles of Spirulina samples were evaluated.

## 2. Results and Discussion

### 2.1. pH Values and Color Coordinates (L*, a* and b*) in the Spirulina Samples

The pH and color coordinates of non-treated and fermented Spirulina samples are given in Appendix A, Table 1 and Figure 1.

When comparing pH values in SMF samples after 24 h, the highest pH (on average, 5.19 and 5.44) was found in samples fermented with *Lactobacillus curvatus* No. 51 and *Lacticaseibacillus paracasei* No. 244; however, most of the pH values were lower than 4.98. After 48 h of SMF, the lowest pH values were attained with samples fermented with *Leuconostoc mesenteroides* No. 225 (4.69). In two (out of ten) SMF samples, the pH values increased after 48 h of fermentation (viz., samples fermented with *Lacticaseibacillus paracasei* No. 244 and *Liquorilactobacillus uvarum* No. 245 strains). However, the same trends were not found in the above mentioned strains when employing SSF.

In 24 h SSF, the pH values of 4 out of 10 analyzed samples were lower in comparison with SMF samples at the same time of fermentation (SSF with *Lactiplantibacillus plantarum* No. 122, *Lacticaseibacillus paracasei* No. 244, *Pediococcus acidilactici* No. 29 and *Liquorilactobacillus uvarum* No. 245). Furthermore, the pH values of two samples were statistically similar in SSF and SMF (viz., fermentation with *Lacticaseibacillus casei* No. 210 and with *Lactobacillus curvatus* No. 51). In the remaining 4 out of 10 analyzed 24 h SSF samples, the pH values were significantly higher in comparison with 24 h SMF samples (i.e., SSF with *Lactobacillus coryniformis* No. 71, *Pediococcus pentosaceus* No. 183, *Levilactobacillus brevis* No. 173, and *Leuconostoc mesenteroides* No. 225).

Among SSF samples after 48 h of fermentation, the lowest pH (4.10) was reached with *Levilactobacillus brevis* No. 173 strain. The increase in acidity is an indicator of the fermentation process and can be affected by many environmental and processing factors, such as LAB strains, carbon sources, type of fermentation, etc. In this respect, SSF are more effective in achieving lower pH values of the fermentable substrate when compared to SMF [40]. It was reported that the greatest changes during the fermentation of Spirulina with the *Lactiplantibacillus plantarum* strain are observed in the first 24 h (pH decreased from 7.3 to 5.1 and remained at the same value after 48 h) [41]. Moreover, Bao et al. [42] reported that the pH values of all fermented Spirulina samples were similar, and significant pH decreases to 4.3–5.3 were observed within the first 12 h. However, the acidification rate was the fastest when fermenting with *Lactiplantibacillus. plantarum* B7 strain [42]. Our study showed that the fermentation conditions (i.e., SMF or SSF) is a key factor on the final pH of the Spirulina samples (*p* = 0.042) (Table 1); in addition, after 48 h of fermentation, most of the pH values decreased in comparison with 24 h of fermentation.

Comparing the L* coordinates in SMF, in most of the cases, SMF samples showed similar L* values as the control (I) samples—except for 24 h SMF with *Lacticaseibacillus paracasei* No. 244, in which L* coordinate was 2.6% higher, and for 24 h SMF with *Pediococcus acidilactici* No. 29 and *Leuconostoc mesenteroides* No. 225, in which L* coordinate was 3.0% lower. Different trends of the L* coordinate were found in SSF samples after 24 and 48 h of fermentation. L* coordinate values were the same as that of the control (II) in 7 out of 20 samples after 24 and 48 h of SSF, chiefly 48 h SSF with *Lactiplantibacillus plantarum* No. 122, 48 h SSF with *Lacticaseibacillus casei* No. 210, 24 and 48 h SSF with *Lactobacillus coryniformis* No. 71, 24 h SSF with *Pediococcus pentosaceus* No. 183, 24 h SSF with *Levilactobacillus brevis* No. 173, and 48 h SSF with *Pediococcus acidilactici* No. 29). Furthermore, L* coordinates were higher than that of the control (II) in 4 out of 20 samples after 24 and 48 h SSF, chiefly 24 h SSF with *Lactiplantibacillus plantarum* No. 122, 48 h SSF with *Lactobacillus curvatus* No. 51, 24 h SSF with *Pediococcus acidilactici* No. 29, and 24 h SSF with *Liquorilactobacillus uvarum* No. 245. Finally, L* coordinates were lower than that of the control (II) in 9 out of 20 samples after 24 and 48 h SSF, viz. 24 h SSF with *Lacticaseibacillus casei* No. 210, 24 h SSF with *Lactobacillus curvatus* No. 51, 24 and 48 h SSF with *Lacticaseibacillus paracasei* No. 244, 48 h SSF with *Pediococcus pentosaceus* No. 183, 48 h SSF with *Levilactobacillus brevis* No. 173, 24 and 48 h SSF with *Leuconostoc mesenteroides* No. 225, and 48 h SSF with *Liquorilactobacillus uvarum* No. 245.

Analysis of between-subject effects unveiled that the analyzed factors and their interactions were not significant concerning the L* coordinates of the samples (Table 1). Additionally, pH and L* coordinate values between samples presented a weak negative correlation (r = −0.277, *p* = 0.002). In contrast to L* coordinates, the LAB strain used for fermentation, duration of fermentation, SMF or SSF conditions, as well as all the interaction of these factors, were statistically significant with respect to a* coordinate values of Spirulina samples (Table 1).

Comparing the a* coordinate of SMF samples with the control (I), lower values were found in 14 out of 20 samples. A similar trend occurred in most of the SSF (with 24 and 48 h fermentation) samples (17 out of 20 samples) in comparison with the control (II). Regarding b* coordinate, most of the SMF samples presented lower values in comparison with the control (I), except for 24 h SMF with *Levilactobacillus brevis* No. 173. Opposite trends were found in b* coordinates of SSF samples, where in all cases, they were higher in comparison with the control (II). The color changes may occur because of the acidification of the substrate medium. Organic acids influence oxidation processes in fermentable substrates, leading to color changes [40]. Spirulina contains different-colored compounds, including carotenoids and C-phycocyanin [43]. In addition, β-cryptoxanthin and zeaxanthin are present in small amounts in Spirulina [44]. It was reported that the phycocyanin molecule is sensitive to environmental conditions, including pH [45,46]. As predicted, it was reported that the L* value of Spirulina has a significant correlation with pigment content [43]. Finally, our study showed that all the analyzed factors and their interactions were significant to the a* coordinate of Spirulina, and these findings led us to conclude that during the fermentation process, changes in Spirulina pigments occur.

### 2.2. l-Glutamic Acid (L-Glu) and Gamma-Aminobutyric Acid (GABA) Concentration in the Spirulina Samples

l-Glutamic acid and gamma-aminobutyric acid concentrations of the non-treated (non-fermented) and fermented Spirulina samples are given in Appendix A, Table 2 and Figure 2.

From the comparison of glutamic acid concentration between 24 h SMF samples and control (I) samples, one may conclude that glutamic acid concentration: was lower in 3 out of 10 samples (in 24 h SMF with *Lactiplantibacillus plantarum* No. 122, *Lacticaseibacillus paracasei* No. 244, and *Leuconostoc mesenteroides* No. 225, by 8.5, 93.8 and 90.3%, respectively); was higher in 6 out of 10 samples (in 24 h SMF with *Lactobacillus curvatus* No. 51, *Lactobacillus coryniformis* No. 71, *Pediococcus pentosaceus* No. 183, *Pediococcus pentosaceus* No. 183, *Pediococcus acidilactici* No. 29, and *Liquorilactobacillus uvarum* No. 245, by 16.9, 7.72, 36.7, 47.2, 18.8 and 46.8%, respectively); and 1 out of 10 samples of glutamic acid concentration was similar (in 24 h SMF with *Lacticaseibacillus casei* No. 210). However, after 48 h of SMF, glutamic acid concentration was found to be higher in 7 out of 10 samples, and it was lower in 3 out of 10 samples when compared to the control (I).

When analyzing glutamic acid concentration SSF samples with the control (II), in most of the cases (except for in 24 and 48 h SSF with *Lacticaseibacillus paracasei* No. 244 samples), glutamic acid concentration increased, and the conditions of fermentation (SMF or SSF) proved to be a statistically significant factor on the glutamic acid concentration in Spirulina samples (Table 2).

Regarding GABA concentration in SMF samples, in all the cases (i.e., after 24 and 48 h of SMF), the values increased consistently in comparison with the control (I), and the highest GABA concentration (286.5 mg/kg) was found in 48 h SMF with *Lacticaseibacillus paracasei* No. 244. The same trend was established in SSF samples, i.e., after 24 and 48 h of SSF, GABA concentration was higher than in the control (II), and the highest concentration of GABA (2395.9 mg/kg) was found in 48 h SSF with *Lacticaseibacillus paracasei* No. 244.

Despite that correlations between glutamic acid and GABA concentrations were not found, it was determined that the LAB strain used for fermentation, the conditions of fermentation (submerged or solid state) as well as the interactions between LAB strain used for fermentation and the conditions of fermentation (submerged or solid state) significantly affected GABA concentration in Spirulina samples. It was reported that LAB strains may produce glutamic acid [47]. Even though the main aim of this study was to evaluate the chemical changes in Spirulina biomass, it can be hypothesized that the correlation between glutamic acid and GABA was not found due to the characteristics of the studied LAB, for which the metabolic pathways include not only the decarboxylation of amino acids but also the production of glutamic acid. Nevertheless, further studies are needed to confirm this hypothesis.

The most common amino acids in *Spirulina* spp. are glutamic acid followed by leucine and aspartic acid [48]. Specific bacterial genera are involved in the production of GABA [49]. It was reported that LAB may induce the structural breakdown of cyanobacterial cell walls via hydrolysis, leading to the conversion of complex compounds [50]. Most of the glutamic acid and GABA-producing microorganisms are LAB, including species from the genera *Lactococcus*, *Lactobacillus*, *Enterococcus* and *Streptococcus* [51]. However, the production of glutamic acid and GABA can vary in relation with microorganism characteristics, and it is species-dependent [52]. In addition to these findings, our study showed that fermentation conditions (SMF or SSF) are also a very statistically significant factor, especially for GABA content in Spirulina samples.

### 2.3. Biogenic Amine (BA) Content in the Spirulina Samples

Biogenic amine (BA) contents in non-treated and fermented Spirulina samples are given in Appendix A, Table 3 and Figure 3 and Figure 4.

Phenylethylamine was not found in Spirulina samples. Furthermore, cadaverine (CAD) and histamine (HIS) were detected in 8 and 5 (out of 42) samples, respectively. However, concentrations of CAD and HIS in 48 h SSF with *Lacticaseibacillus paracasei* No. 244 were 24.4 and 137.9 mg/kg, respectively. In addition, concentrations of CAD higher than 30 mg/kg were found in 24 h SSF with *Lacticaseibacillus casei* No. 210; nevertheless, CAD was not detected after 48 h of SSF in the same samples. All the analyzed factors and their interactions were statistically significant regarding the concentration of CAD and HIS in Spirulina samples (*p* ≤ 0.0001) (Appendix A).

With respect to tryptamine (TRP) concentration, it was not found in most of the SMF samples (except for in 24 h SMF *Lacticaseibacillus paracasei* No. 244). Still, TRP was found in all the SSF samples, and its content ranged from 2.66 mg/kg (in 24 h SSF with *Lactobacillus curvatus* No. 51 samples) to 7.53 mg/kg (in 48 h SSF with *Liquorilactobacillus uvarum* No. 245 samples).

Putrescine (PUT) and spermidine (SPRMD) were the main BA found in fermented Spirulina samples. Concerning PUT, higher PUT concentrations were found in SSF samples in most of the cases, in comparison with SMF ones, and the highest PUT content was obtained in SSF with *Levilactobacillus brevis* No. 173 samples (833.4 mg/kg after 24 h and 854.7 mg/kg after 48 h). The lowest PUT concentration was found in samples fermented with *Pediococcus pentosaceus* No. 183, viz., in 24 h SMF and 24 h SSF, PUT concentrations were 0 and 20.8 mg/kg, respectively; in 48 h SMF and 48 h SSF, PUT concentrations were 86.6 and 32.4 mg/kg, respectively. All the analyzed factors and their interactions were statistically significant regarding the concentration of PUT in Spirulina samples (*p* ≤ 0.0001) (Appendix A).

In regard to SPRMD in SMF samples, it was found that SMF decreased SPRMD concentration in Spirulina samples, on average from 3.3 to 4.9 times (in 48 h SMF with *Lacticaseibacillus paracasei* No. 244 and in 24 h SMF with *Pediococcus pentosaceus* No. 183 samples, respectively). However, opposite trends of the SPRMD concentration were found in SSF samples. Furthermore SPRMD concentrations were, on average, 7.7 times higher in SSF than in SMF samples. Additionally, all the analyzed factors and most of their interactions—except for the interaction between duration of fermentation and conditions of fermentation (submerged or solid state)—were statistically significant on SPRMD concentration in Spirulina samples (*p* ≤ 0.0001) (Appendix A), and the highest SPRMD concentration was obtained in 24 h SSF with *Lactobacillus curvatus* No. 51 and 24 and 48 h SSF with *Levilactobacillus brevis* No. 173 (on average, 519.9 mg/kg).

When observing the results of tyramine (TYR), the concentration in most of the SMF samples was lower than in the control (I), except in SMF with *Lacticaseibacillus paracasei* No. 244, *Levilactobacillus brevis* No. 173 and in *Leuconostoc mesenteroides* No. 225 and *Liquorilactobacillus uvarum* No. 245. Conversely, opposite trends were perceived in SSF samples, i.e., all the SSF samples exhibited higher TYR concentration when compared to the control (II). The highest TYR concentration was found in SSF samples with *Lacticaseibacillus paracasei* No. 244 (TYR concentration was 167.7 mg/kg after 24 h and 609.4 mg/kg after 48 h SSF). Furthermore, all the analyzed factors and their interactions were statistically significant regarding TYR concentration in Spirulina samples (*p* ≤ 0.0001) (Appendix A).

The comparison of SPRM in Spirulina samples with control (I) samples led us to conclude that SMF reduced its concentration. However, in SSF samples, SPRM concentration was similar or slightly higher in comparison to the control (II) samples (except for 48 h SSF with *Leuconostoc mesenteroides* No. 225 samples). All the analyzed factors and their interactions were statistically significant regarding their effect on SPRM concentration in Spirulina samples (*p* ≤ 0.05) (Appendix A).

Changes in eating habits and looking for functional compounds are often associated with the incorporation of new non-traditional food ingredients into the main diet. However, functional properties in most of these so-called “super foods” are causing concern in terms of food safety issues. For this reason, this study includes not only the evaluation of GABA but also of BA concentrations in fermented Spirulina samples due to the fact that both compounds are formed through the carboxylation of amino acids. Table 3 tabulates the correlations between BA and GABA and glutamic acid concentrations in Spirulina samples. More specifically, from these results, it was possible to unfold statistically significant correlations between TRP, PUT, CAD, HIS, TYR and SPRMD and GABA, as well as between TRP, PUT, SPRMD and SPRM and glutamic acid. TYR, HIS, PUT, CAD, SPRM and SPRMD are mainly produced by microbial decarboxylation of amino acids [53,54]. PUT is a precursor for the synthesis of SPRMD [54]. Likewise, PUT and CAD can be metabolized from ornithine and lysine, respectively [55]. TYR is associated with constricting of vascular system, and HIS is known as a vasodilator [56]. In addition to the individual toxicity of BA, Wang et al. [57] reported that the sum of primary, secondary and tertiary biogenic amines is very important. TYR causes migraines, and PUT and CAD potentiate intoxication in the presence of other BA [58]. Finally, the samples in which the highest GABA concentrations were found also presented the highest content of BA. This shows that it is important to simultaneously study the presence of functional and non-desirable compounds in the end-product, especially when both compounds (as in this case) are produced through decarboxylation pathways of amino acids. Figure 4 presents the principal component analysis (PCA) of the first two principal components (PC) and makes apparent the existence of two clusters formed by the SMF and SSF samples, respectively, and thus the existence of statistically significant differences between both type of fermentations. Our previous studies showed that during the SSF, microorganisms show more efficient capacity to excrete enzymes and to degrade fermentable substrates [59]. 

### 2.4. Fatty Acid (FA) Profile in the Spirulina Samples

Fatty acid (FA) content in non-treated and fermented Spirulina samples is given in Appendix A and Figure 5.

The main FA in non-treated and fermented Spirulina samples were methyl palmitate (C16:0), methyl linoleate (C18:2) and gamma-linolenic acid methyl ester (C18:3ɣ). When investigating C16:0 content in non-treated and SMF samples, the concentration was higher in 18 out of 20 samples than that in the control (I) samples. Only in the cases of 24 and 48 h SMF with *Lacticaseibacillus casei* No. 210 samples was the concentration of C16:0 similar to the control (I) (on average, 42.4% from total fat content). The same investigation of C16:0 content but in the SSF samples led to discovering different trends, and the highest C16:0 concentration was obtained in 48 h SSF with *Lactobacillus coryniformis* No. 71 samples (on average, 60.8% from total fat content). Tests between subjects showed that the LAB strain used for fermentation, the interaction LAB strain used for fermentation ∗ duration of fermentation and the interaction LAB strain used for fermentation ∗ conditions of fermentation (submerged or solid state) were statistically different in terms of C16:0 concentration in Spirulina samples (Appendix A).

Regarding the content of C18:2 in Spirulina samples, the values were lower in all the SMF and SSF samples in comparison with the control (I) and control (II) samples, respectively. Tests between subjects showed that the C18:2 content in Spirulina samples was significantly affected in the following cases: LAB strain used for fermentation (*p* = 0.003); conditions of fermentation (SMF or SSF) (*p* = 0.038); LAB strain used for fermentation ∗ duration of fermentation (*p* ≤ 0.0001); LAB strain used for fermentation ∗ conditions of fermentation (SMF or SSF) (*p* ≤ 0.001); and LAB strain used for fermentation ∗ duration of fermentation ∗ conditions of fermentation (SMF or SSF) (*p* ≤ 0.0001) (Appendix A).

When comparing the content of C18:3ɣ with the control, different trends in SMF and SSF samples were perceived. In the case of SMF, C18:3ɣ content increased in 6 out of 20 SMF samples; C18:3ɣ content decreased in 8 out of 20 SMF samples; and C18:3ɣ content remained similar to the control (I) in 6 out of 20 SMF samples. In the case of SSF, C18:3ɣ content increased in 14 out of 20 SSF samples; C18:3ɣ content decreased in 4 out of 20 SSF samples; and C18:3ɣ content remained similar to the control (II) in 2 out of 20 SSF samples. Tests between subjects showed that the C18:3ɣ content in Spirulina samples was significantly affected in the following cases: LAB strain used for fermentation (*p* = 0.004); LAB strain used for fermentation ∗ duration of fermentation (*p* ≤ 0.0001); LAB strain used for fermentation ∗ conditions of fermentation (SMF or SSF) (*p* ≤ 0.001); and LAB strain used for fermentation ∗ duration of fermentation ∗ conditions of fermentation (SMF or SSF) (*p* ≤ 0.0001) (Appendix A).

Alfa-linolenic acid methyl ester (C18:3α) was only found in SSF samples, and its content ranged from 0.399 (in 48 h SSF with *Liquorilactobacillus uvarum* No. 245 samples) to 0.618% of total fat content (in 24 h SSF with *Leuconostoc mesenteroides* No. 225 samples). Tests between subjects showed that all the analyzed factors were significant regarding the concentration of C18:3α in Spirulina samples (*p* ≤ 0.0001) (Appendix A).

Methyl palmitoleate (C16:1), methyl stearate (C18:0), and *cis, trans*-9-oleic acid methyl ester (C18:1 *cis, trans*) contents in Spirulina samples were lower than 5% from the total fat content. In addition, the analyzed factors and their interaction proved to not be significant on C16:1 content in Spirulina samples. On the other hand, the LAB strain used for fermentation, the conditions of fermentation (SMF or SSF), and the interaction LAB strain used for fermentation ∗ conditions of fermentation (SMF or SSF) were significant regarding C16:1 content in Spirulina samples. Likewise, the interaction LAB strain used for fermentation ∗ duration of fermentation was significant on C18:1 *cis, trans* in Spirulina samples.

The proximate composition of spirulina is related to numerous factors such as the source of the cyanobacteria, the season of the year, as well as to the manufacturing technology. The lipid concentration of *Arthrospira platensis* can vary from ca. 5 to 10% (of the dry weight) [60]. Long-chain FA are predominant compounds in Spirulina (mainly palmitic acid and gamma-linoleic acid) [61,62]. However, other studies reported higher contents of palmitic (46%), oleic (8%) and linoleic (12%) acids in Spirulina and lower contents of gamma-linoleic acid (20%) and stearic acid (1%) [63]. One of the most significant polyunsaturated FA is gamma-linoleic acid [62,64]. In addition to the FA profile of non-treated Spirulina, it was reported that 6 days of SSF with the fungus *Aspergillus niger*, *Spirulina* spp. attained the highest concentration of linoleic acid (60.63%, from total fat content), which was significantly higher than that obtained by SSF with *Lactiplantibacillus plantarum* (16.93%). However, the contents of elaidic, α-linoleic, stearic and palmitic acids of *Spirulina* spp. were higher in SSF with *Lactiplantibacillus plantarum*. The desirable changes in FA profile were explained by reduction of the substrate concentration during the fermentation process, because the nutrients were used for microbial growth and secondary metabolite production [65].

Omega-6 constitutes the majority of the total Spirulina FA [66,67]. Furthermore, Spirulina contains a significant amount of palmitic acid (16:0), which represents more than 25% from the total fat content [60]. PUFA levels in Spirulina ranged from 1.5 to 2.0% of total fat [68], whereas PUFA content represented 30% of the total fat content [69]. Another study reported that the FA profile of Spirulina contains sapienic acid (2.25 mg/100 g), linoleic acid (16.7%) and *γ*-linolenic acid (14%) [70]. According to Liestianty et al. [71], the FA of Spirulina encompasses myristic, heptadecanoic, stearic, oleic, palmitoleic, omega-3, omega-6, linoleic and palmitic acids. According to Al-Dhabi and Valan Arasu [70], myristic, stearic and eicosadienoic acids were the predominant saturated FA in Spirulina. Spirulina is the only food source that contains large amounts of essential FA, especially *γ*-linolenic acid. Finally, the FA profile of Spirulina samples is highly dependent on the fermentation process; thus, by selecting the most appropriate pre-treatment conditions desirable, changes in the FA profiles may be achieved.

## 3. Conclusions

All the tested LAB strains were suitable for Spirulina fermentation, and the lowest pH value (4.10) was obtained after 48 h of SSF with *Levilactobacillus brevis* No. 173. Changes in the pigments of Spirulina occurred during the fermentation process, and all the analyzed factors and their interactions were significant regarding the color’s Spirulina a* coordinate. The main FA in non-treated and fermented Spirulina samples were methyl palmitate (C16:0), methyl linoleate (C18:2) and gamma-linolenic (C18:3ɣ) acid methyl esters. Likewise, changes in the FA profile of the Spirulina were detected throughout the fermentation processes. Moreover, fermentation increased glutamic acid and GABA concentrations in Spirulina samples, and the highest GABA concentration was found in 48 h SMF with *Lacticaseibacillus paracasei* No. 244 (286.5 mg/kg) and in 48 h SSF with *Lacticaseibacillus paracasei* No. 244 (2395.9 mg/kg). Furthermore, putrescine (PUT) and spermidine (SPRMD) were the main BA in fermented Spirulina samples. In addition, significant correlations were found between BA concentration and GABA and glutamic acid. Spirulina samples where the highest GABA concentrations were found also showed the highest content of BA. Such correlation underlines the importance to study not only functional compounds but also potentially undesirable substances simultaneously, especially when they are involved in similar decarboxylation pathways of the amino acids.

## 4. Materials and Methods

### 4.1. Spirulina and Lactic Acid Bacteria Strains Used in Experiments and Fermentation Conditions

Lyophilized Spirulina powder (*Arthrospira platensis*) (content per 100 g: sodium 1.1 g, total carbohydrates 30.3 g, proteins 60.6 g, calcium 151.5 mg, potassium 1.7 mg, iron 48.5 mg) was provided by Now Foods Company (Bloomingdale, IL, USA).

The LAB strains (Lactiplantibacillus plantarum No. 122; Lacticaseibacillus casei No. 210; Lactobacillus curvatus No. 51; Lacticaseibacillus paracasei No. 244; Lactobacillus coryniformis No. 71; Pediococcus pentosaceus No. 183; Levilactobacillus brevis No. 173; Pediococcus acidilactici No. 29; Leuconostoc mesenteroides No. 225; Liquorilactobacillus uvarum No. 245) were acquired from the Lithuanian University of Health Sciences collection (Kaunas, Lithuania). Before the experiment, LAB strains were incubated and multiplied in De Man, Rogosa, and Sharpe (MRS) broth culture medium (Biolife, Milano, Italy) at 30 °C under anaerobic conditions for 24 h. A total of 3 mL of fresh LAB grown in MRS broth (average cell concentration of 9.0 log_10_ CFU/mL) was inoculated in 100 mL of Spirulina media (for SMF, Spirulina powder was mixed with sterilized water, in a ratio of 1:20 *w*/*w*, whereas for the SSF Spirulina/water, the ratio was 1:2 *w*/*w*)—thus giving rise to 3% (*v*/*w*) of purified LAB strain per Spirulina–water mixture.

Afterward, the algae samples were fermented under anaerobic conditions in a chamber incubator (Memmert GmbH Co. KG, Schwabach, Germany) for 24 and 48 h, at 30 °C. Non-fermented samples (mixed with sterilized water in appropriate proportions for SMF and SSF) were analyzed as a control. Before and after fermentation, the pH, color coordinates, glutamic acid, GABA, BA and FA concentrations of the samples were analyzed. The experimental design is schematized in Figure 6.

### 4.2. Analysis of pH and Color Coordinates (L*, a* and b*) in the Spirulina Samples

The pH of Spirulina samples was evaluated with a pH meter (Inolab 3, Hanna Instruments, Venet, Italy) by inserting the pH electrode into the algae samples. The color coordinates of the Spirulina samples were evaluated on the surface using the CIE L*a*b* system (CromaMeter CR-400, Konica Minolta, Marunouchi, Tokyo, Japan) [72].

### 4.3. Evaluation of l-Glutamic Acid (L-Glu) and Gamma-Aminobutyric Acid (GABA) Concentration in Spirulina Samples

First, 0.5 g of seaweed samples was extracted in 50 mL of Milli-Q water for 10 min using an overhead shaker. The samples were incubated for 30 min at 60 °C in a water bath. Then, the tubes were cooled down and centrifuged at 4500 rpm for 10 min. A 1 mL aliquot of the supernatant was transferred into 15 mL polypropylene test tubes and diluted with 9 mL of Milli-Q water. Finally, samples were filtered and transferred to a 2 mL autosampler vial. Analysis was performed on a TSQ Quantiva MS/MS coupled to Thermo Scientific Ultimate 3000 HPLC instrument (Thermo Scientific, Waltham, MA, USA). Chromatographic separation was carried out on a Luna Omega Polar C18 (2.1 × 100 mm, 3.0 μm) column at 40 °C using an injection volume of 5 μL. The mobile phase consisted of a 0,5 mM ammonium acetate solution in Milli-Q water (eluent A) and methanol (eluent B). A flow rate of 0.2 mL/min was used. The following gradient conditions were applied: 0.00 min, 1% B (99% A); 1.00 min, 1.0% B (99% A); 6.00 min, 99% B (1% A); 7.50 min, 99% B (1% A); 8.00 min, 1% B (99% A); 10.00 min, 1% B (99% A). LC-MS interface conditions for the ionization of GABA and L-Glu in the positive ESI mode were as follows: needle voltage + 4500 V; sheath gas 60 Arb; aux gas 25 Arb; sweep gas 5 Arb; ion transfer tube temperature 200 °C; vaporizer temperature 350 °C. The main fragments were identified using the selected reaction monitoring (SRM), with the following ionic transitions: GABA (*m*/*z* 104 > *m*/*z* 45.151, CE 25.72 V; *m*/*z* 104 > 69.165, CE 15.92 V; *m*/*z* 104 > *m*/*z* 87.36, CE 10.66 V); L-Glu (*m*/*z* 148 > *m*/*z* 56.05, CE 30 V; *m*/*z* 148 > *m*/*z* 84, CE 30 V). Method recovery ranged from 59% to 112% for GABA and from 58% to 152% for L-Glu. Method repeatability ranged from 5% to 23% for GABA and from 1% to 20% for L-Glu. The results were obtained in some rounds of experiments on different days.

### 4.4. Analysis of Biogenic Amine (BA) Concentration in the Spirulina Samples

Sample preparation and determination of the BAs, including tryptamine (TRP), phenylethylamine (PHE), putrescine (PUT), cadaverine (CAD), histamine (HIS), tyramine (TYR), spermidine (SPRMD) and spermine (SPRM), in Spirulina samples was conducted by following the procedure reported by Ben-Gigirey et al. [73] with some modifications. Briefly, the standard BA solutions were prepared by dissolving known amounts of each BA (including internal standard) in 20 mL of deionized water. The extraction of BA in samples (5 g) was performed by using 0.4 mol/L perchloric acid. The derivatization of sample extracts and standards was performed using dansyl chloride solution (10 mg/mL) as a reagent. The chromatographic analyses were carried out using a Varian ProStar HPLC system (Varian Corp., Palo Alto, CA, USA) with two ProStar 210 pumps, a ProStar 410 auto-sampler, a ProStar 325 UV/VIS Detector and Galaxy software (Agilent, Santa Clara, CA, USA) for data processing. For the separation of amines, a Discovery ^®^ HS C18 column (150 × 4.6 mm, 5 μm; SupelcoTM Analytical, Bellefonte, PA, USA) was used. The eluents were ammonium acetate (A) and acetonitrile (B), and the elution program consisted of a gradient system with a 0.8 mL/min flow-rate. The detection wavelength was set to 254 nm, the oven temperature was 40 °C, and samples were injected in 20 μL aliquots. The target compounds were identified based on their retention times in comparison to their corresponding standards.

### 4.5. Analysis of Fatty Acid (FA) Profile in the Spirulina Samples

The extraction of lipids for fatty acids (FA) analysis was performed with chloroform/methanol (2:1 *v*/*v*), and FA methyl esters (FAME) were prepared according to Pérez-Palacios et al. [74]. The fatty acid composition of the Spirulina samples was identified using a gas chromatograph GC-2010 Plus (Shimadzu Europa GmbH, Duisburg, Germany) equipped with Mass Spectrometer GCMS-QP2010 (Shimadzu Europa GmbH, Duisburg, Germany). Separation was carried out on a Stabilwax-MS column (30 m length, 0.25 mmID, and 0.25 μm df) (Restek Corporation, Bellefonte, PA, USA). Oven temperature program started at 50 °C, then increased at a rate of 8 °C/min to 220 °C, held for 1 min at 220 °C, increased again at a rate of 20 °C/min to 240 °C and, finally, held throughout 10 min. The injector temperature was 240 °C, interface −240 °C, and ion source 240 °C. The carrier gas was helium at a flow-rate of 0.91 mL/min. The individual FAME peaks were identified by comparing their retention times with FAME standards (Merck & Co., Inc., Kenilworth, NJ, USA).

### 4.6. Statistical Analysis

Fermentation of the samples was performed in duplicate, and all analytical experiments were carried out in triplicate. To evaluate a potential influence of different factors (SMF or SMF conditions, duration of fermentation, type of LAB strain used for fermentation) and their interaction on Spirulina sample characteristics, the mean of values was calculated, using the statistical package SPSS for Windows (v28.0.1.0 (142), SPSS, Chicago, IL, USA), and was compared using Duncan’s multiple range test with significance defined at *p* ≤ 0.05. A linear Pearson’s correlation was used to quantify the strength of the relationship between the variables. The results were recognized as statistically significant at *p* ≤ 0.05. 

## Figures and Tables

**Figure 1 toxins-15-00075-f001:**
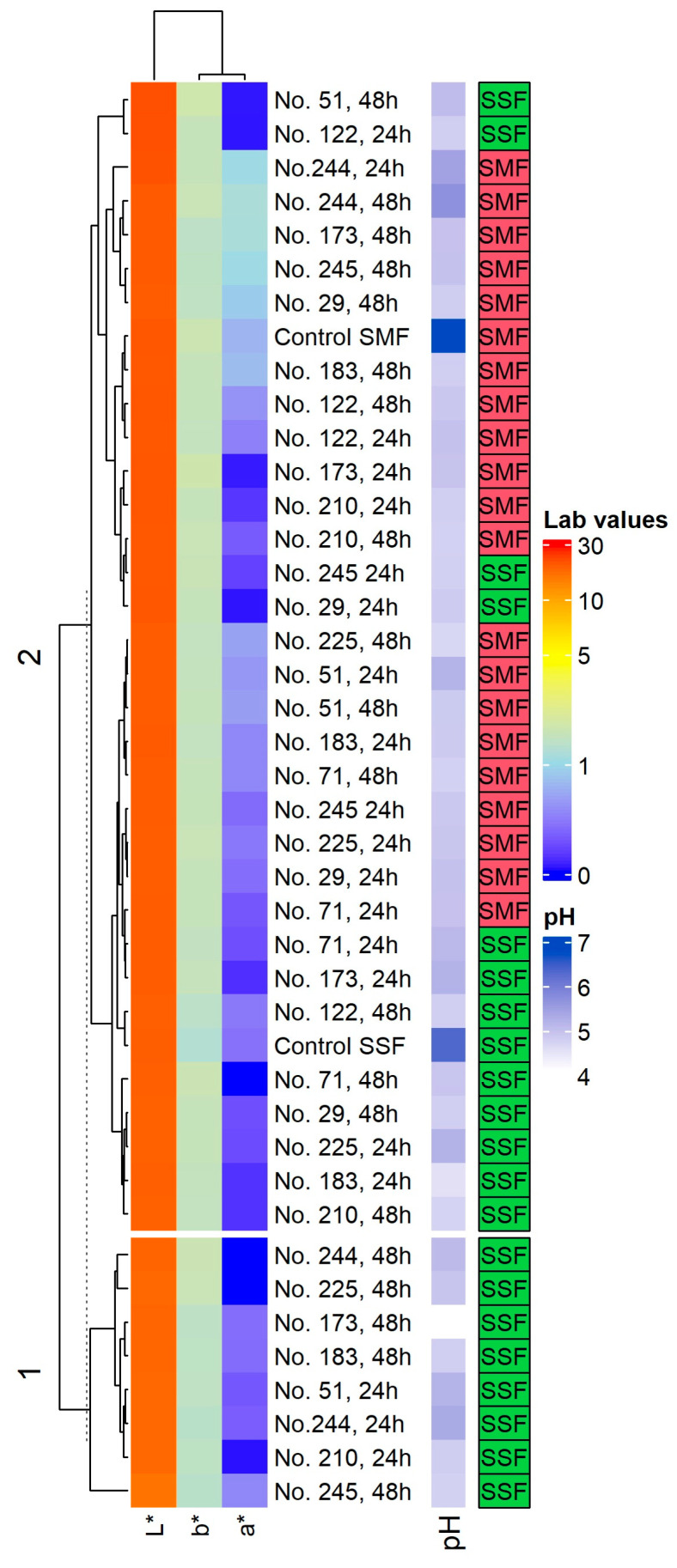
Changes in pH values and color coordinates (L*, a* and b*) in non-treated and fermented Spirulina samples. Note: No. 122—*Lactiplantibacillus plantarum*; No. 210—*Lacticaseibacillus casei*; No. 51—*Lactobacillus curvatus*; No. 244—*Lacticaseibacillus paracasei*; No. 71—*Lactobacillus coryniformis*; No. 183—*Pediococcus pentosaceus*; No. 173—*Levilactobacillus brevis*; No. 29—*Pediococcus acidilactici*; No. 225—*Leuconostoc mesenteroides*; No. 245—*Liquorilactobacillus uvarum*; SMF—submerged fermentation; SSF—solid state fermentation; L*, a and b* coordinates for the color in the CIE L*a*b* system.

**Figure 2 toxins-15-00075-f002:**
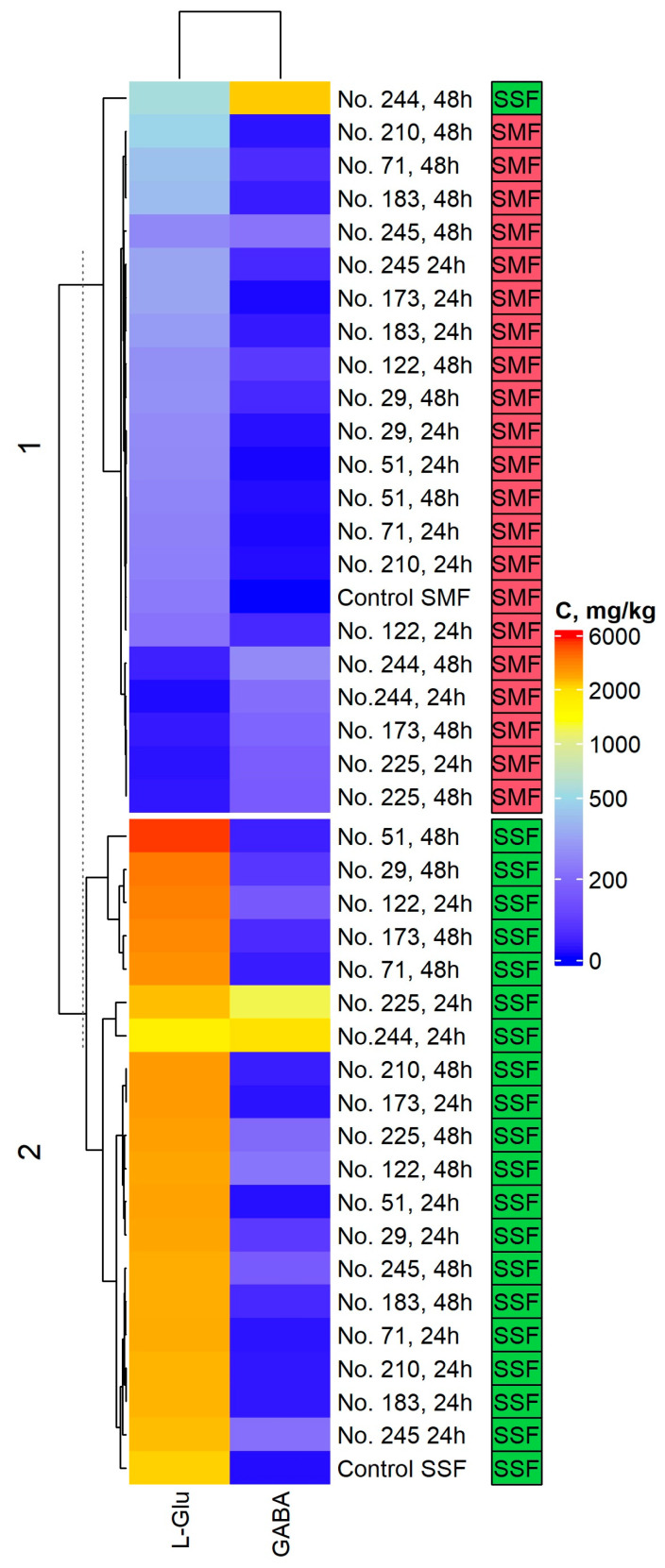
Changes in l-glutamic acid (l-Glu) and gamma-aminobutyric acid (GABA) concentrations in non-treated and fermented Spirulina samples. Note: No. 122—*Lactiplantibacillus plantarum*; No. 210—*Lacticaseibacillus casei*; No. 51—*Lactobacillus curvatus*; No. 244—*Lacticaseibacillus paracasei*; No. 71—*Lactobacillus coryniformis*; No. 183—*Pediococcus pentosaceus*; No. 173—*Levilactobacillus brevis*; No. 29—*Pediococcus acidilactici*; No. 225—*Leuconostoc mesenteroides*; No. 245—*Liquorilactobacillus uvarum*; SMF—submerged fermentation; SSF—solid state fermentation; C—concentration in mg/kg.

**Figure 3 toxins-15-00075-f003:**
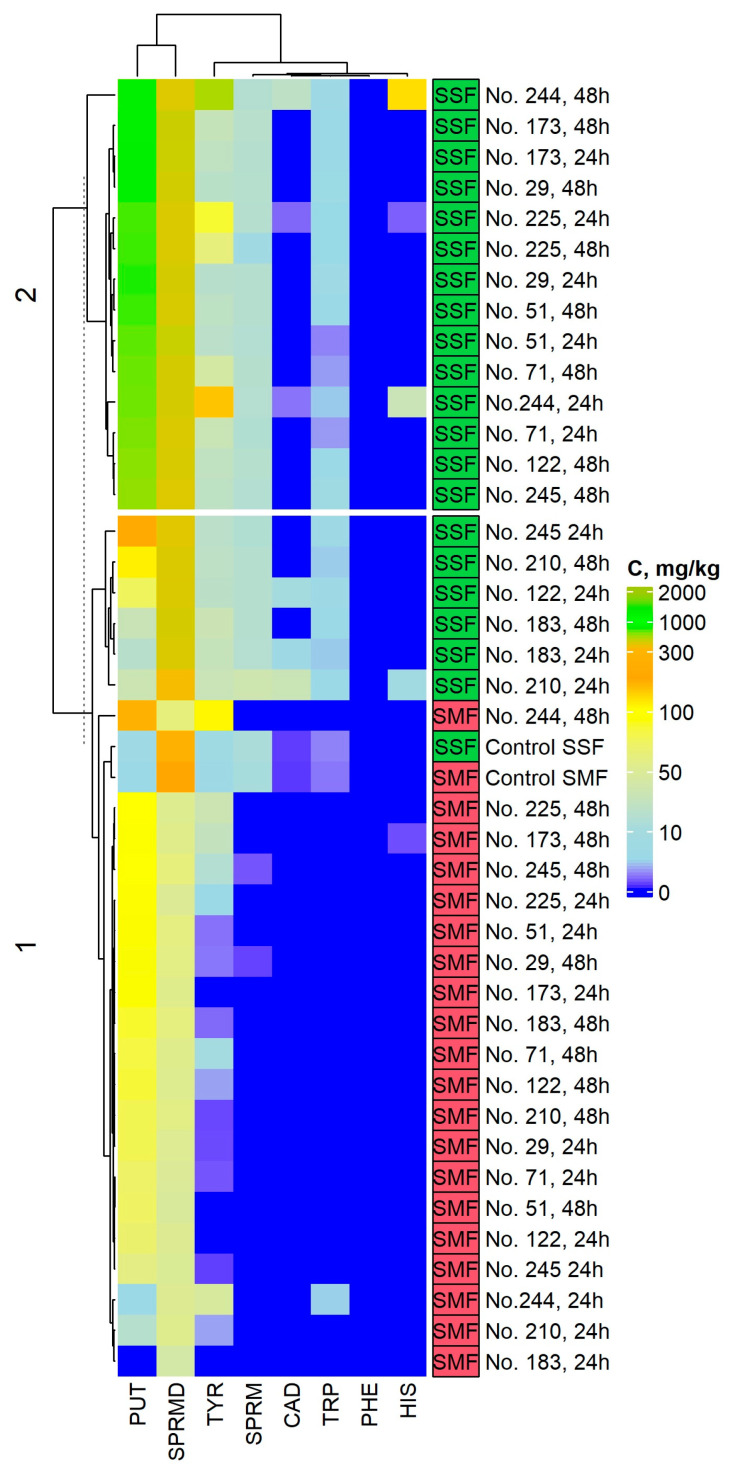
Changes in biogenic amine (BA) concentrations in non-treated and fermented Spirulina samples. Note PUT – putrescine; TRP—tryptamine; PHE—phenylethylamine; CAD—cadaverine; HIS—histamine; TYR—tyramine; SPRMD—spermidine; SPRM—spermine; No. 122—*Lactiplantibacillus plantarum*; No. 210—*Lacticaseibacillus casei*; No. 51—*Lactobacillus curvatus*; No. 244—*Lacticaseibacillus paracasei*; No. 71—*Lactobacillus coryniformis*; No. 183—*Pediococcus pentosaceus*; No. 173—*Levilactobacillus brevis*; No. 29—*Pediococcus acidilactici*; No. 225—*Leuconostoc mesenteroides*; No. 245—*Liquorilactobacillus uvarum*; SMF—submerged fermentation; SSF—solid state fermentation; C—concentration in mg/kg.

**Figure 4 toxins-15-00075-f004:**
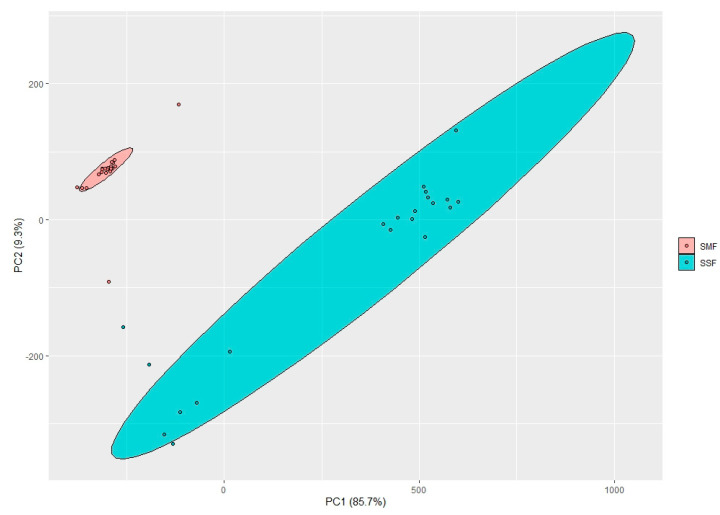
Biogenic amines distribution in submerged (SMF) and solid-state fermented (SSF) samples. Projection of the variables (biogenic amines) in the principal components 1 (PC1) and 2 (PC2), obtained by principal components analysis (PCA). The percentage of variability accounted for by PC1 and PC2 is 85.7% and 9.3%, respectively.

**Figure 5 toxins-15-00075-f005:**
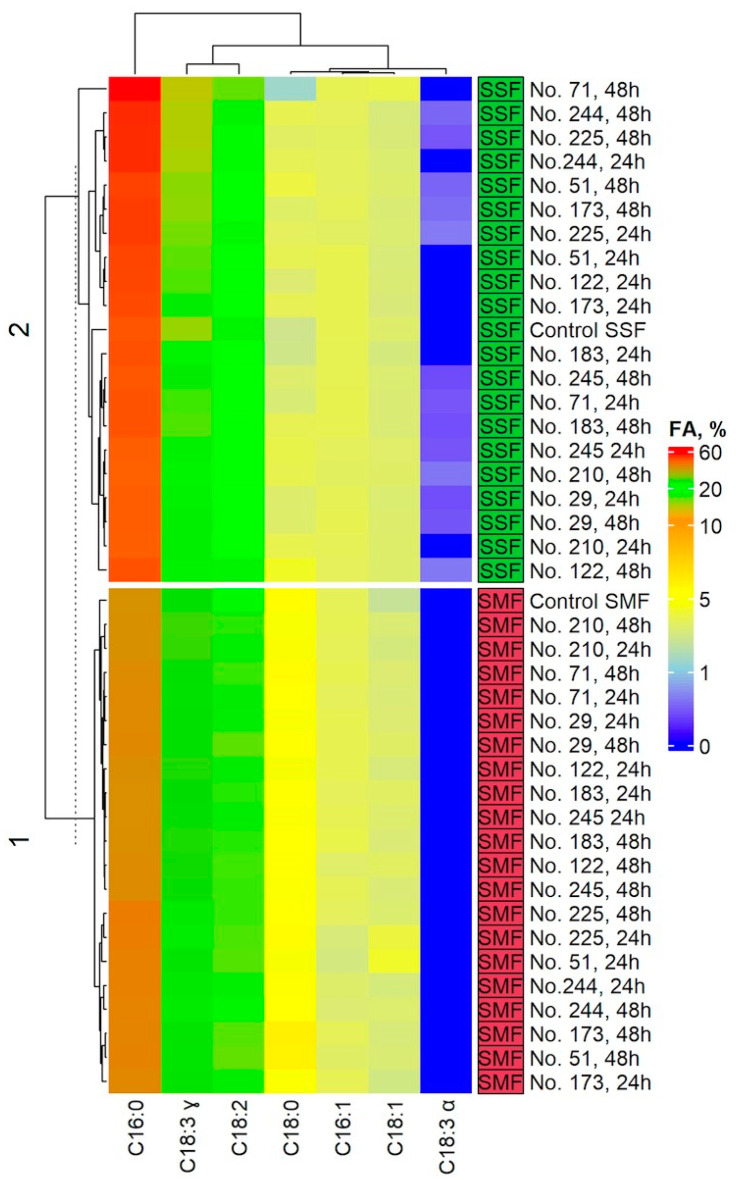
Changes in fatty acid (FA) profile in non-treated and fermented Spirulina samples. Note: C16:0—methyl palmitate; C16:1—methyl palmitoleate; C18:0—methyl stearate; C18:1 *cis, trans*—*cis, trans*-9- oleic acid methyl ester; C18:2—methyl linoleate; C18:3ɣ—gamma- linolenic acid methyl ester; C18:3α—alfa linolenic acid methyl ester; No. 122—*Lactiplantibacillus plantarum*; No. 210—*Lacticaseibacillus casei*; No. 51—*Lactobacillus curvatus*; No. 244—*Lacticaseibacillus paracasei*; No. 71—*Lactobacillus coryniformis*; No. 183—*Pediococcus pentosaceus*; No. 173—*Levilactobacillus brevis*; No. 29—*Pediococcus acidilactici*; No. 225—*Leuconostoc mesenteroides*; No. 245—*Liquorilactobacillus uvarum*; SMF—submerged fermentation; SSF—solid state fermentation; % from total fat content.

**Figure 6 toxins-15-00075-f006:**
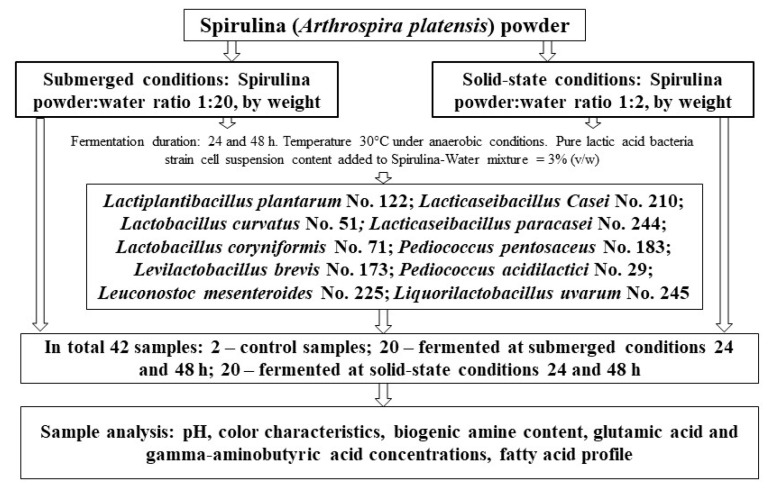
Schematic representation of the experimental design in this research study.

**Table 1 toxins-15-00075-t001:** Influence of the analyzed factors and their interaction on the color coordinates (L*, a* and b*) and pH values of Spirulina samples.

Factors and Their Interaction	Dependent Variable	*p*
Lactic acid bacteria strain used for fermentation	L*	0.403
	a*	**0.0001**
	b*	0.377
	pH	0.791
Duration of fermentation	L*	0.317
	a*	**0.0001**
	b*	0.807
	pH	0.898
Conditions of fermentation (submerged or solid state)	L*	0.438
	a*	**0.0001**
	b*	0.286
	pH	**0.042**
Lactic acid bacteria strain used for fermentation × Duration of fermentation	L*	0.422
	a*	**0.0001**
	b*	0.448
	pH	0.719
Lactic acid bacteria strain used for fermentation × Conditions of fermentation (submerged or solid state)	L*	0.398
	a*	**0.0001**
	b*	0.112
	pH	0.439
Duration of fermentation × Conditions of fermentation (submerged or solid state)	L*	0.307
	a*	**0.012**
	b*	0.313
	pH	0.665
Lactic acid bacteria strain used for fermentation × Duration of fermentation × Conditions of fermentation (submerged or solid state)	L*	0.393
	a*	**0.0001**
	b*	0.197
	pH	0.486

L*—lightness; a*—redness or −a*—greenness; b*—yellowness; −b*—blueness; influence of factor or factors interaction is recognized as statistically significant when *p* ≤ 0.05. Significant influence of the analyzed factors or their interactions are marked in bold letters.

**Table 2 toxins-15-00075-t002:** Influence of the analyzed factors and their interactions on L-glutamic acid (L-Glu) and gamma-aminobutyric acid (GABA) concentration in Spirulina samples.

Factors and Their Interaction	Dependent Variable	*p*
Lactic acid bacteria strain used for fermentation	GABA	**0.0001**
	l-Glutamic acid	0.641
Duration of fermentation	GABA	0.987
	l-Glutamic acid	0.328
Conditions of fermentation (submerged or solid state)	GABA	**0.020**
	l-Glutamic acid	**0.0001**
Lactic acid bacteria strain used for fermentation × Duration of fermentation	GABA	0.813
	l-Glutamic acid	0.942
Lactic acid bacteria strain used for fermentation × Conditions of fermentation (submerged or solid state)	GABA	**0.0001**
	l-Glutamic acid	0.740
Duration of fermentation × Conditions of fermentation (submerged or solid state)	GABA	0.499
	l-Glutamic acid	0.358
Lactic acid bacteria strain used for fermentation × Duration of fermentation × Conditions of fermentation (submerged or solid state)	GABA	0.893
	l-Glutamic acid	0.957

Gamma-aminobutyric acid (GABA); influence of factor or factors interaction is recognized as statistically significant when *p* ≤ 0.05. Significant influences of the analyzed factors or their interactions are marked in bold letters.

**Table 3 toxins-15-00075-t003:** Correlations between biogenic amines and gamma-aminobutyric acid (GABA) and L-glutamic acid (L-Glu) concentrations.

Correlations	Correlation (r)	Significance (p)	Correlation (r)	Significance (p)
	with GABA		with l-Glutamic acid	
TRP	0.215 *	0.016	0.541 **	0.0001
PUT	0.309 **	0.0001	0.486 **	0.0001
CAD	0.298 **	0.001	−0.076	0.401
HIS	0.648 **	0.0001	−0.073	0.414
TYR	0.681 **	0.0001	−0.014	0.879
SPRMD	0.211 *	0.018	0.627 **	0.0001
SPRM	0.172	0.054	0.528 **	0.0001

TRP—tryptamine; PUT—putrescine; CAD—cadaverine; HIS—histamine; TYR—tyramine; SPRMD—spermidine; SPRM—spermine. ** Correlation is significant at the 0.01 level (2-tailed). * Correlation is significant at the 0.05 level (2-tailed).

## Data Availability

The data are available from the corresponding author upon reasonable request.

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
