# Peer review of "Changes in Spirulina’s Physical and Chemical Properties during Submerged and Solid-State Lacto-Fermentation"

_toxins, 2023, doi:10.3390/toxins15010075_

Round 1

Reviewer 1 Report

The authors have studied The changes in Spirulina physical and chemical properties dur ing the submerged and solid-state lacto-fermentation. Numerous and extensive Results part. Please see my few suggestions regarding this paper:

In the Introduction I suggest a strong paragraph regarding Spirulina. Information about it is missing. Data about Spirulina should be provide so as any reader of the manuscript to understand its role, importance and effects. What it is? what it is used for? what it is administered for? data on human studies? data on experimental studies? (for experimental, I suggest checking https://doi.org/10.1007/s11356-018-2761-0  and https://doi.org/10.1007/s11356-019-04249-4

L86-95. What is the novelty this study brings to the field? This aspect must also better be highlighted in the Key contribution section, page 1 of the manuscript.

Why so many results are provided in the Supplementary material/files and not in the main manuscript? It is remarkable the amount of results presented as supplementary files.

Author Response

Reviewer 1: The authors have studied The changes in Spirulina physical and chemical properties dur ing the submerged and solid-state lacto-fermentation. Numerous and extensive Results part.

Authors response: Authors are very thankful for valuable comments.

Reviewer 1: Please see my few suggestions regarding this paper:

In the Introduction I suggest a strong paragraph regarding Spirulina. Information about it is missing. Data about Spirulina should be provide so as any reader of the manuscript to understand its role, importance and effects. What it is? what it is used for? what it is administered for? data on human studies? data on experimental studies? (for experimental, I suggest checking https://doi.org/10.1007/s11356-018-2761-0  and https://doi.org/10.1007/s11356-019-04249-4

Authors response: Authors are thankful for comment, suggested information was included.

Reviewer 1: L86-95. What is the novelty this study brings to the field? This aspect must also better be highlighted in the Key contribution section, page 1 of the manuscript.

Authors response: Authors are thankful for valuable comment, corrected:

The main finding, which is highlighted in this study, is, that the samples in which the highest GABA concentrations were found, also dis-played the highest content of BA. For this reason, not only the concentration of functional compounds in the end-product must be controlled, but also non-desirable substances, because both of these compounds are produced through similar metabolic pathways of the decarboxylation of amino acids.

Reviewer 1: Why so many results are provided in the Supplementary material/files and not in the main manuscript? It is remarkable the amount of results presented as supplementary files.

Authors response: Authors are thankful for comment. We would like to explain, that in Supplementary material concretized data in numbers are given, however in manuscript visualization of the results is given, where numbers are not seen exactly.

Reviewer 2 Report

The manuscript shows interesting results related to the changes in physical properties and formation of functional compositions from Spirulina using different fermentation conditions and lactic acid bacteria strains

An acceptance for publishing the manuscript on the journal is recommended because the manuscript exhibits novel and significant results in the research field. The writing was well prepared and the experiments were well design.  Some minor revisions are suggested to improve the quality of the manuscript:

1.      Line 46-50: please remove unnecessary references. Only the most related references should be cited.

2.      Line 61: Please replace “Nowadays” by another word.

3.      The data description is too detailed and thereby sometime leading to confusion to the readers. The data description just needs to emphasize the most important observed data/changes/findings. It is not necessary to show all the changing trends or compared all kinds of data.

4.      Line 333: Please change “Figure 5” to “Figure 4” and some more analysis/explanation for the PCA results should be added to solidify the concluding statement.

Author Response

Reviewer 2: The manuscript shows interesting results related to the changes in physical properties and formation of functional compositions from Spirulina using different fermentation conditions and lactic acid bacteria strains

An acceptance for publishing the manuscript on the journal is recommended because the manuscript exhibits novel and significant results in the research field. The writing was well prepared and the experiments were well design.  Some minor revisions are suggested to improve the quality of the manuscript.

Authors response: Authors are thankful for valuable comments.

Reviewer 2:

  1. Line 46-50: please remove unnecessary references. Only the most related references should be cited.

Authors response: Authors are thankful for comment, the most related references were cited.

Reviewer 2:

  1. Line 61: Please replace “Nowadays” by another word.

Authors response: corrected.

Reviewer 2:

  1. The data description is too detailed and thereby sometime leading to confusion to the readers. The data description just needs to emphasize the most important observed data/changes/findings. It is not necessary to show all the changing trends or compared all kinds of data.

Authors response: Authors are thankful for comment. We would like to explain, that the data were compared, by comparing groups of the samples after 24 and 48 hours of SMF and SSF. We would like to leave such style of description, because thera are just the most important results highlighted.

Reviewer 2:

  1. Line 333: Please change “Figure 5” to “Figure 4” and some more analysis/explanation for the PCA results should be added to solidify the concluding statement.

Authors response: Authors are thankful for valuable comment, an explanation was included:

Figure 4 present the principal component analysis (PCA) of the first two principal components (PC) and makes apparent the existence of 2 clusters formed by the SMF and SSF samples, respectively, thus the existence of statistically significant differences between both type of fermentations. Our previous studies showed, that during the SSF, microorganisms are showing more efficient capacity to excrete enzymes and to degrade fermentable substrate [59].
